# Efficacy of an Irritable Bowel Syndrome Diet in the Treatment of Small Intestinal Bacterial Overgrowth: A Narrative Review

**DOI:** 10.3390/nu14163382

**Published:** 2022-08-17

**Authors:** Justyna Paulina Wielgosz-Grochowska, Nicole Domanski, Małgorzata Ewa Drywień

**Affiliations:** 1Department of Human Nutrition, Institute of Human Nutrition Sciences, Warsaw University of Life Sciences, 02-776 Warsaw, Poland; 2Faculty of Pharmaceutical Sciences, University of British Columbia, Vancouver, BC V6T 1Z4, Canada

**Keywords:** microbiota, dysbiosis, IBS, SIBO, FODMAP, probiotics, prebiotics, fiber

## Abstract

Small intestinal bacterial overgrowth (SIBO) is highly prevalent in irritable bowel syndrome (IBS). The eradication of bacterial overgrowth with antibiotics is the first-line treatment. However, focusing only on the antimicrobial effects without taking care to improve lifestyle factors, especially dietary patterns, may predispose patients to intestinal microbiota dysfunction. The objective of this study is to determine whether the current recommendations regarding nutrition in IBS are suitable for patients with SIBO. A narrative literature review was carried out using databases, including PubMed, ScienceDirect and Google Scholar. Recent studies indicate that dietary manipulation may have a role in alleviating SIBO gastrointestinal symptoms. A low FODMAP diet proposed for IBS may promote a negative shift in the gut microbiota and deepen the existing state of dysbiosis in SIBO patients. Supplementation with soluble fiber can lessen the symptoms in IBS and SIBO. Targeted probiotic therapy may also increase the effectiveness of antibiotic treatment and regulate bowel movements. Therefore, optimal dietary patterns play a key role in the treatment of SIBO. Based on currently available literature, the potential efficacy of the IBS diet in SIBO is largely hypothetical. Future research is needed to characterize a specific diet for the treatment of SIBO.

## 1. Introduction

The gut microbiota represents an essential human organ with a variety of beneficial functions for the host. The human body is inhabited by microbes, including bacteria, viruses, fungi, archaea and protozoa [1]. It is well established that the gastrointestinal (GI) tract has the largest microbial colonization, where over 70% of the body’s microorganisms reside in the large intestine [2]. Nonetheless, the composition of the gut microbiota differs between individuals and is influenced by various external factors. Recent studies have demonstrated that a richer, more diverse and balanced gut microbiota composition is linked to a healthier and longer life [3,4]. However, disturbances in the quantity and/or quality of gut microbiota, defined as dysbiosis, may predispose individuals to an increased risk of certain chronic diseases, as well the development of small intestinal bacterial overgrowth (SIBO) [5,6].

The diagnosis of SIBO can be performed via invasive or noninvasive methods. In either case, a confirmed diagnosis is recognized by abnormal amounts of a bacterial or methanogenic load [7]. To distinguish between the three types of SIBO (methane-dominant, hydrogen-dominant and sulfide-dominant), it is necessary to measure the type of gas produced by the microorganism during the fermentation process and evaluate it via breath tests if required [8].

SIBO is characterized by nonspecific GI symptoms, similar to irritable bowel syndrome (IBS), whereby changes in bowel movements are seen [9]. Gut dysbiosis has also been presented in IBS patients [10]. It has been shown that up to 78% of patients with IBS also test positive for SIBO [2]. However, symptoms of SIBO are often neglected and are attributed to other underlying diseases of the patient. Ignoring the existence of SIBO may lead to disturbances in the metabolism of carbohydrates, proteins, fats and vitamins, as well as to changes in the amount of digestive enzymes produced. Failure to properly treat SIBO can result in weight loss and inflammation throughout the body [11,12].

Although SIBO has been well studied and the eradication of bacterial overgrowth with antibiotics is recognized as an effective first-line treatment, some patients are resistant to therapy. Moreover, SIBO recurrence has been seen in 43% patients at 9 months after completing antibiotic treatment [13]. Hence, focusing only on the antimicrobial effects without taking care to improve lifestyle factors, especially dietary patterns, may not yield satisfactory results and may even predispose further GI dysfunction. Since the clinical picture of patients with SIBO and IBS is comparable, it might be suggested that the recommendations for IBS patients can be applied to patients with SIBO, although this has yet to be established.

Fortunately, the optimal nutritional management of IBS has already been well documented [14]. This includes implementing a low-FODMAP diet, supplementing with probiotics and prebiotics, improving lifestyle factors and avoiding harmful products that may aggravate IBS symptoms [15,16]. A low-FODMAP diet is based on the elimination of foods high in fermentable oligo-, di- and monosaccharides, and polyols (FODMAPs), which may lessen symptoms, such abdominal pain and distention, bloating, constipation and diarrhea, and improve quality of life in IBS patients. Symptomatic relief is the result of decreased intestinal osmotic activity and a reduction in gas production from the bacterial fermentation of unabsorbed, undigested carbohydrates in the colon [17,18]. Probiotics, specifically single-strain monoprobiotics, and dietary fiber also have shown enormous potential to regulate gut motility and improve symptoms in IBS patients [19,20].

However, little is known about the effect of these therapeutic dietary interventions and the optimal diet and lifestyle for SIBO patients. The objective of this study is to determine whether the current recommendations regarding nutrition in IBS would be suitable for patients with SIBO.

## 2. Materials and Methods

A comprehensive search of PubMed, ScienceDirect and Google Scholar was conducted from 2012 to 2022 to identify suitable literature. The search strategy included the following terms: (low FODMAP OR high FODMAP)AND (clinical trial OR randomized controlled trial OR cross-sectional study OR crossover study OR retrospective study OR intervention study) AND (IBS OR irritable bowel syndrome) AND (SIBO OR small intestinal bacterial overgrowth) AND (probiotic OR monoprobiotic OR bacterial strain) AND (fiber OR soluble fiber OR psyllium OR inulin OR phgg) AND (mmc OR migrating motor complex) AND (mindful eating or mindfulness training). The abstracts and titles of relevant articles were screened and only full-length papers with quantitative statistical analyses were included in this narrative review. Conference proceedings, abstracts and meta-analyses were not included. Studies based on children, pregnant women, animals, in vitro experiments, poliprobiotic treatments or studies reported in non-English languages were excluded. Studies with results on inflammatory bowel disease, ulcerative colitis, colon cancer, colorectal cancer or celiac disease were also excluded. Ultimately, the review was limited to studies conducted in patients with SIBO, IBS and functional gastrointestinal disorders (FGIDs) and healthy people.

## 3. Results and Discussion

The initial search strategy yielded 65 articles and, following a comprehensive review, 34 studies comprised 25 randomized controlled trials, 5 clinical trials, 2 cross-sectional studies, 1 retrospective study and 1 pilot dietary intervention study, which were analyzed for inclusion into the narrative review. An analysis of these findings was presented under the following four categories: low FODMAP diet, fiber, monoprobiotics and mindful eating. The detailed search process is illustrated in Figure 1.

### 3.1. Low-FODMAP Diet

Altogether, twelve studies identified potential connections between a low-FODMAP diet and the microbiota profile, which are presented in Table 1 [17,21,22,23,24,25,26,27,28,29,30,31]. The majority of papers compared the low-FODMAP diet in IBS patients to traditional dietary recommendations, habitual diets or a high-FODMAP diet. None of the studies evaluated the impact of this diet in SIBO patients. Three studies investigated the relationship between probiotics or prebiotics in combination with the low-FODMAP diet on the gut microbiota [21,25,29]. Nine authors proposed that the implementation of a low-FODMAP diet for 4 to 9 weeks may adversely decrease the abundance of *Actinobacteria,* especially beneficial *Bifidobacterium* [21,22,23,24,25,26,27,28,29], while the coadministration of a probiotic and prebiotic (fructo-oligosaccharides (FOSs), but not B-galactooligosaccharides (B-GOSs)) with the low-FODMAP diet may reverse these changes [21,25,29]. Contrary to those studies, one paper found an inverse correlation, suggesting an increase in the quantity of *Bifidobacterium* and *Lactobacillus;* however, this was following a low FODMAP and gluten-free diet [31]. Other studies showed that a reduction in FODMAP foods can significantly increase saccharolytic *Bacteroides, Porphyromonadaceae* and nonsaccharolytic taxon *Bilophilia* [23,24,28]. *Bilophilia*, a hydrogen-sulfide-producing species, may likely be involved in the pathogenesis of hydrogen sulfide SIBO. Halmos et al. [26] suggested an average reduction of 47% in the total bacterial load following adherence to the low-FODMAP diet, with a negative reduction in *Akkermansia muciniphila*, which is involved in enhancing host metabolic functions and *Faecalibacterium prausnitzii*, which may have anti-inflammatory properties. Furthermore, a study conducted by Bennet et al. [27] revealed that 42% of IBS patients scored higher on the dysbiosis index (DI) after 4 weeks on the low-FODMAP diet, while a traditional diet decreased DI scores in 33% of IBS patients. However, four authors did not find a reduction in the diversity of microbiota, including one study which lasted 6 months [23,26,28,30]. Only two of the included studies focused on measuring exhaled gases in breath tests. McIntosh et al. [28] demonstrated only a slight depletion in hydrogen production in the low FODMAP group compared with the high FODMAP group, whereas Patcharatrakul et al. [17] noticed that postprandial hydrogen breath production was significantly lower in the low FODMAP group in comparison to a commonly recommended diet. A low-FODMAP diet might improve the well-being of patients with GI symptoms; however, long-term dietary adherence may have negative health effects. According to one study, a restriction in FODMAP foods decreased symptoms in 86% of patients with IBS [18]. Conversely, many of the included studies postulated that a prolonged restriction of FODMAP foods may be linked to undesirable alterations in gut microbiota, with similar results observed by other authors [32,33,34]. FODMAPs act as prebiotics and positively modulate the microbiome by stimulating the growth of *Akkermansia municiphila, Bifidobacteria*, and *Faecalibacterium prausnitzii* and promoting the production of short-chain fatty acids (SCFAs) [35]. Therefore, a low-FODMAP diet might be characterized as antiprebiotic because of the reduction in beneficial bacteria species [29]. In SIBO, the treatment priority, apart from addressing risk factors and identifying the underlying cause, should be striving for the state of eubiosis, characterized by the balance of microbiota colonization. Hence, it remains uncertain whether a low-FODMAP diet is helpful or necessary for patients with SIBO, especially for prolonged periods of time.

### 3.2. Monoprobiotics

The impact of monostrain probiotic supplementation on the modification of gut microbiota was assessed in eleven studies and is demonstrated in Table 2 and Table 3 [36,37,38,39,40,41,42,43,44,45,46]. Only one study by García-Collinot et al. [39] included patients with SIBO. The results of this paper concluded that supplementation with *Sacharomyces boulardii* (CNCM I 745) in SIBO patients with systemic sclerosis was associated with significantly higher eradication rates and a decline in exhaled hydrogen, as compared to metronidazole therapy alone [39]. The nine remaining studies were carried out in patients with IBS [37,42,43,44,45,46], constipation [36,38,41] and healthy individuals [40]. Hydrogen or methane breath tests were only measured in three papers [39,40,41]. Only two-week supplementation with *Bifidobacterium Infantis* 35624 increased methane production in the lactulose breath test (LBT) and led to positive SIBO tests in some subjects. In contrast, four-week *Lactobacillus reuterii* (DSM 17938) administration significantly decreased methane production, with complete methane disappearance (<5 ppm at LBT) observed in 55% subjects. The administration of *Bifidobacterium Infantis* and *Lactobacillus reuterii* found no association with changes in hydrogen production [40,41]. In eight studies, authors observed significant GI symptom relief with *B. coagulans* LBSC (DSM17654) [37], *B. coagulans* (MTCC 5856) [42], *B. coagulans Unique* IS [36], *L. plantarum* 299v (DSM 9843) [43], *S. cerevisiae* CNCM I-3856 [44], *S. boulardii* CNCM I 745 [45] and *B. animalis subsp. lactis* BB-12 [38] following 4–8 weeks of treatment. Eight-week supplementation with *S. cerevisiae* CNCM I-significantly reduced abdominal pain scores and improved stool consistency after only four weeks of treatment in the IBS-M, IBS-C and IBS-D subgroups [44]. A similar effect was observed after intervention with *B. coagulans* LBSC (DSM17654) [37], *B. coagulans* (MTCC 5856) [42], *B. coagulans Unique* IS [36], *L. plantarum* 299v (DSM 9843) [43], *B. animalis subsp. lactis* and BB-12 [38]. Interestingly, no significant difference was noticed after administering a higher dose (ten billion rather than one billion) of probiotic BB-12 on defecation frequency [38]. A three-week administration of *S. boulardii* CNCM I 745 led to considerable differences regarding diarrhea, flatulence, gurgling, gas release, eructation and pain severity [47]. Symptoms associated with diarrhea also improved after implementation with *Lactobacillus reuterii* (DSM 17938) [41], *B. coagulans* LBSC (DSM17654) [37] and *B. coagulans* (MTCC 5856) [42]. Furthermore, supplementation with *L. paracasei* HA-196 or *B. longum* R0175 appeared to ameliorate the quality of life based on the IBS-QOL score [31]. In particular, the *L. paracasei* group required fewer rescue medications in comparison with the placebo group [46].

Probiotic therapy may provide effective relief of bloating, abdominal pain, diarrhea, flatulence and may improve stool consistency. However, probiotics are strain-dependent—hence, there is a need for the precise, individualized selection of probiotic strains based on their specific properties and the intended effect during supplementation [48]. Based on the included studies, monoprobiotics with well characterized strains may have a favorable role in preventing the progression of IBS and SIBO symptoms; however, evidence is still lacking in SIBO patients. To date, only one comprehensive meta-analysis and systematic review has explored probiotic treatment in SIBO, and many of the included papers in this meta-analysis focused on poliprobiotics and did not provide a full classification of the strain types [49]. There is a necessity to focus on common probiotic strains, such as *Sacharomyces boulardii* (CNCM I 745) and *Lactobacillus reuterii* (DSM 17938), with eradication properties that might be essential to SIBO therapy. Every effort should be under taken to select bacterial strains dedicated to each specific type of SIBO, which would not worsen the patients’ symptoms or promote overgrowth.

### 3.3. Fiber

In total, seven studies analyzed the association between fiber supplementation and the impact on the gut microbiome [20,50,51,52,53,54,55]. Table 4 provides detailed information about the characteristics of the included studies. Dietary supplementation with soluble fiber was related to positive changes in the bacterial composition of the gut microbiota [20,52,54,55]. The implementation of psyllium for 7 days in constipated subjects resulted in significant increases in beneficial microorganisms, such as *Faecalibacterium*, *Lachnospira* and *Roseburia*. These are connected with producing SCFAs such as butyrate and increased fecal water absorption [20]. Holscher et al. [54] demonstrated that adding agave inulin to healthy adult diets improved gut microbiota diversity, including a reduction in *Desulfovibrio* and an increase in *Actinobacteria* and *Bifidobacteria*. In another randomized control trial, the combination of partially hydrolyzed guar gum (PHGG) and inulin for 3 weeks significantly decreased Clostridium sp. [55]. Moreover, four authors observed that adding psyllium husk or PHGG to a regular diet may improve IBS symptoms, such as abdominal pain, bloating or gasses, as well as improve stool consistency and frequency [20,51,52,53]. Switching from a high-fiber diet to a low-fiber diet (<11 g/1000 cal) in 16 healthy volunteers for 7 days was associated with the development of GI symptoms in every participant of the study. Moreover, SIBO was diagnosed in two subjects after this short-term intervention with a low-fiber diet [50].

Similar results were found with other authors [56,57]. Garg [57] concluded that the intake of 25 g of psyllium husk with 500 mL of water for 12 weeks resulted in a major relief of IBS symptoms. However, Oskouie et al. [56] presented that IBS was more prevalent in individuals with a low intake of dietary fiber.

Dietary fiber should be considered an essential nutrient for the growth of beneficial microorganisms with prebiotic potential. The included studies support that increasing the intake of fiber, in particular, soluble fiber, may yield satisfactory results in patients with GI symptoms and modulate gut microbiota; however, studies in SIBO patients are still needed.

**Table 4 nutrients-14-03382-t004:** Characteristics of included studies connected with fiber.

Author, Year Type of Study	Period	Study Group	Intervention/Control	Methods
Saffouri et al., 2019 [50] Pilot interventional study	7 days	16 healthy	<11 g fiber/ 1000 cal/day	Breath test
Jalanka et al.,2019 [20] Randomized, controlled trail	7 days	16 constipated 8 healthy	21 g/day Psyllium husk/ maltodextrin	16S rRNA
Reider et al., 2020 [52] Clinical trial	9 weeks	20 healthy	5g PHGG/ 3 time per day	16S rRNA
Holscher et al., 2015 [54] Randomized, controlled trail	21 days	29 healthy	5.0g/7.5 g/0.0g/day agave inulin	16S rRNA
Linetzky et al., 2012 [55] Randomized clinical trial	3 weeks	60 constipated	15g/day inulin+ PHGG, maltodextrin	PCR
Niv et al., 2016 [51] Randomized clinical trial	18 weeks	121 IBS	6g PHGG group/ placebo	Francis severity IBS score
Polymeros et al., 2013 [53] Uncontrolled open-label trial	4 weeks	49 chronic constipated	5 mg PHGG/day	Bristol stool scale

PHGG— partially hydrolyzed guar gum

### 3.4. Mindful Eating

The migrating motor complex (MMC) acts as a “gastrointestinal keeper”, and is responsible for cleansing the GI tract from food debris and sweeping excess bacteria into the colon [58]. In one paper, patients with SIBO were reported to have a lower frequency of phase III MMCs, thus, acknowledging it as a risk factor for SIBO [58]. Hence, mindful eating, defined as appropriate breaks between meals, including the omission of snacking, might be a key element in the prevention and treatment of SIBO.

Based on four included studies [59,60,61,62] presented in Table 5, the results demonstrated associations between dietary patterns, the prevalence of IBS and functional dyspepsia. In one randomized controlled trial, participants who were not paying attention to sufficient chewing had a higher risk of IBS [59]. According to Zaribaf et al. [61], subjects who demonstrated meal irregularity tended to experience problems with frequent and severe abdominal pain. A lower probability of developing functional diarrhea, functional constipation and IBS was related to slower eating rates during lunch and effective food chewing [62]. Interestingly, a randomized controlled trial comparing the effectiveness of a 4-week low-FODMAP diet vs. education on when and how to eat, rather than what to eat, yielded similar results with a decrease in IBS symptoms [60].

These studies suggest that proper dietary patters could be a key element in the prevention of functional GI disorders, however, the results have not been replicated in the SIBO population.

## 4. Strengths and Limitations

To our knowledge, this was the first narrative review to investigate the efficacy of an IBS diet for the treatment of SIBO. This review emphasized the negative effects of prolonged low-FODMAP diets, which may disturb and shift microbiota composition and worsen the existing state of dysbiosis in SIBO patients. Moreover, this was the first review including only monoprobiotics strains, which demonstrated a need for the careful selection of bacterial species with proper nomenclature, and their importance in achieving specific clinical effects in IBS subtypes or potentially SIBO. Our review also raised the topic of adding adequate amounts of fiber, especially soluble fiber, to a regular diet as an important element in the modulation of gut microbiota and a reduction in clinical symptoms. We also deduced that mindful eating, including slower consumption, meal regularity and sufficient chewing, could help decrease the percentage of GI symptoms and cannot be overlooked in therapy. Nevertheless, our narrative review had several limitations. Firstly, due to the lack of studies in the SIBO population, most of the categories reviewed were not investigated in this group of patients. However, the clinical picture and microbiota profile are similar to IBS and SIBO; hence, it was suggested that the response to treatment might be comparable. Another limitation of this review was the studies’ design: the small sample sizes of the study groups [17,20,23,25,26,27,39,40,41,45,46,50,55], short follow-up periods [23,39,51,53] and the lack of control groups [25,41,50,52,53,59]. Only half of the included studies were randomized controlled trials. Furthermore, in some included papers, the placebo effect could not be avoided [22,30,38,46,53]. Lastly, it is unclear how accurately the participants followed the low-FODMAP diet, especially in studies where no meals were provided and the participants were only asked to fill out a food diary, as this likely influenced the results of the breath tests [17,27,60].

## 5. Conclusions

This narrative review suggested that there is a favorable association with monoprobiotics, fiber supplementation and mindful eating, and negative effects associated with low-FODMAP diets on the gut microbiome, especially in IBS patients. Applying these recommendations to the treatment of SIBO was inconclusive due to a lack of research including SIBO patients in the studies. Based on the currently available literature, the potential efficacy of the IBS diet in SIBO is largely hypothetical and future research is needed to characterize the specific dietary recommendations for the treatment of SIBO.

## Figures and Tables

**Figure 1 nutrients-14-03382-f001:**
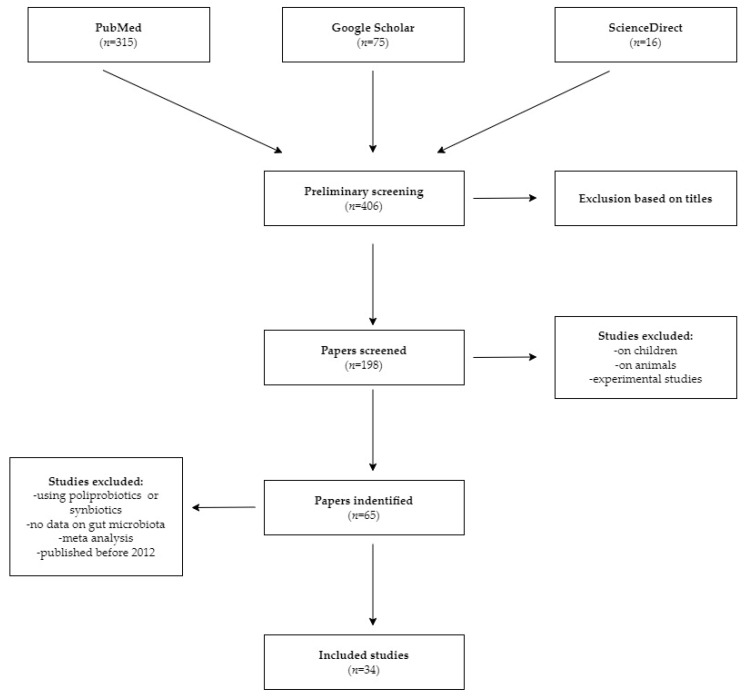
Flow diagram of the records included in the narrative review.

**Table 1 nutrients-14-03382-t001:** Characteristics of included studies connected with a low-FODMAP diet on gut microbiota.

Author, Year, Type of Study	Period	Study Group	Intervention/Control	Methods	Outcome
Patcharatrakul et al., 2019 [17] Randomized controlled trial	4 weeks	62 IBS	SILFD/BRD	Breath test	↓H_2_ volume
McIntoshet al., 2017 [28] Randomized controlled trial	3 weeks	37 IBS	LFD/HFD	Breath test 16S rRNA	↓H_2_ volume ↓*Bifidobacterium* ↑*Porphyromonadaceae*
Bennet et al., 2018 [27] Randomized controlled trial	4 weeks	67 IBS	LFD/TDA	GA-map Dysbiosis test	↓*Bifidobacterium*, ↑Dysbiosis index
Zhang et al., 2021 [24] Parallel-group, randomized controlled trial	3 weeks	100 IBS	LFD/TDA	16S rRNA	↓*Bifidobacterium*, ↓*Fusobacterium*, ↓*Bacterioides* ↑*Bilophila*
Huaman et al., 2018 [23] Randomized controlled trial	4 weeks	40 FGIDs	LFD/MD +(B-GOS)	16S rRNA	↓*Bifidobacterium* ↑*Bilophila*
Halmos et al., 2015 [26] Single-blinded, randomized, crossover trial	3 weeks	27 IBS 6 Healthy	LFD/AD	16S rRNA	↓*Akkermansia Muciniphila* ↓*Faecalibacterium prausnitzii* ↓*Ruminococcus torques*
Staudacher et al., 2012 [22] Randomized controlled trial	4 weeks	41 IBS	LFD/HD	FISH	↓*Bifidobacterium*
Naseri et al., 2021 [31] Clinical trial	6 weeks	42 IBS	LF-GFD	16S rRNA	↑*Bacterioides* ↑*Bifidobacterium, Lactobacillus*
Wilson et al., 2020 [29] Randomized placebo-controlled trial	4 weeks	69 IBS	LFD +(B-GOS/placebo)/sham diet + placebo supplement	16S rRNA	↓*Bifidobacterium*, ↓*Actinobacteria*
Staudacher et al., 2021 [21] Randomized controlled trial	4 weeks	95 IBS	LFD/sham diet +(probiotic/placebo)	16S rRNA	↓*Bifidobacterium* ↑*Bacterioides*
Hustoft et al., 2017 [25] Randomized, double- blinded, placebo-controlled crossover study	9 weeks	20 IBS	LFD/HFD + FOS/maltodextrin	16S rRNA	↓*Bifidobacterium*, ↓*Clostridium, Faecalibacterium* ↑*Bilophila*
Harvie et al., 2017 [30] Randomized controlled trial	6 months	50 IBS	LFD/TDA + reintroduction	16S rRNA	No change in the microbiota

LFD—low-FODMAP diet; TDA—traditional dietary advice; LF-GFD—low FODMAP gluten-free diet, SLFD—structural individual low-FODMAP diet; BRD—brief advice on a commonly recommended diet; FGIDs—functional gastrointestinal disorders; ↑—increase; ↓—decrease;

**Table 2 nutrients-14-03382-t002:** Characteristics of included studies connected with monoprobiotics.

Authors, Year, Type of Study	Duration of Study	Study Group	Intervention/Control	Dose/Day
Gayathri et al., 2021 [44] Randomized controlled trial	8 weeks	100 IBS	*S. cerevisiae* CNCM I-3856/placebo	2 × 10^9^ CFU
Gupta et al., 2021 [37] Randomized controlled trial	80 days	40 IBS	*Bacillus coagulans* LBSC (DSM17654)/placebo	2 × 10^12^ CFU
Madempud et al., 2020 [36] Randomized controlled trial	4 weeks	100 FC	*B. coagulans Unique* IS2/placebo	2 × 10^12^ CFU
García-Collinot et al., 2020 [39] Clinical trial	2 months	75 SIBO + SSc	*S.Boulardii*(SB)/metronidazole (M)SB + M	200 mg
Lewis et al., 2020 [46] Randomized controlled trial	8 weeks	251 IBS	*Lactobacillus paracasei* HA-196/*Bifidobacterium longum* R0175/placebo	10 × 10^9^ CFU
Kumar et al., 2018 [40] Randomized controlled trial	2 weeks	19 healthy	*B.infantis* 35624/placebo	No data
Ojetti et al., 2017 [41] Retrospective study	4 weeks	20 constipated	*L. reuteri* (DSM 17938)	2 × 10^8^ CFU
Majeed et al., 2016 [42] Randomized controlled trial	3 months	36 IBS-D	*B. coagulans*(MTCC 5856)/placebo	2 × 10^9^ CFU
Eskesen et al., 2015 [38] Randomized controlled trial	4 weeks	1248 with low defecation frequency	*Bifidobacterium* animalis subsp. lactis, BB-12/placebo	1 × 10^12^ or 10× 10^12^ CFU
Akhondi-Meybodi et al., 2014 [45] Randomized clinical trial	3 weeks	60 IBS	*Saccharomyces boulardii* CNCM I 745/placebo	200mg
Ducrotté et al., 2012 [43] Clinical trial	4 weeks	214 IBS	*L. plantarum* 299v (DSM 9843)/placebo	10 × 10^12^ CFU

FC—functional constipation; IBS-D—diarrhea-predominant IBS; SSc—systemic sclerosis.

**Table 3 nutrients-14-03382-t003:** Characteristics of included studies connected with monoprobiotics.

	Key Results							
Fully Characterized Strains	Diarrhea	Stool Frequency/ Consistency	Bloating	Abdominal Pain	SBM	Gas Release	H2 Volume	CH4 Volume
*B. infantis* 35624	ND ^1^	ND	ND	ND	ND	ND	No change	↑
*L. reuteri* (DSM 17938)	↓	ND	ND	ND	ND	ND	No change	↓
*B. coagulans*(MTCC 5856)	↓	↑	↓	↓	ND	ND	ND	ND
*B. coagulans* LBSC (DSM17654)	↓	↑	↓	↓	ND	ND	ND	ND
*B. coagulans Unique* IS2	ND	↑	ND	↓	ND	ND	ND	ND
*L. plantarum* 299v (DSM 9843)	ND	↑	↓	↓	ND	ND	ND	ND
*S. cerevisiae* CNCM I-3856	ND	↑	ND	↓	ND	ND	ND	ND
*S. boulardii* CNCM I 745	↓	ND	↓	↓	ND	↓	↓	ND
*B. animalis* subsp. lactis, BB-12	ND	↑	↓	↓	ND	ND	ND	ND
*L. paracasei* HA-196	ND	ND	ND	ND	ND	ND	ND	ND

^1^ ND—no data; ↑—increase; ↓—decrease;

**Table 5 nutrients-14-03382-t005:** Characteristics of included studies connected with mindful eating.

Author, Year, Type of Study	Study Group	Characteristics of Group	Methods
Zaribaf et al., 2019 [61] Cross-sectional study	4763 adults	Iranian	Rome III questionnaire Questions about: meal patterns, eating rate, chewing quality
Vakhshuury et al., 2019 [62] Cross-sectional study	600 adults	Military personnel in Iran	Rome III questionnaire, FFQ Questions about: breakfast consumption, lunch intake time, chewing efficiency
Khayyatzadeh et al., 2018 [59] Randomized controlled trial	988 adults	Iranian	Rome III questionnaire, FFQ, dietary behaviors assessment Questions about: meal pattern, chewing quality
Böhn et al., 2015 [60] Randomized controlled trial	75 adults	18–70 year old Swedish met Rome III criteria for IBS	Rome III criteria, IBS-SSS questionnaire 4-day food diary

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
