# Peer review of "Efficacy of an Irritable Bowel Syndrome Diet in the Treatment of Small Intestinal Bacterial Overgrowth: A Narrative Review"

_nutrients, 2022, doi:10.3390/nu14163382_

Round 1
Reviewer 1 Report
The narrative review written by Justyna et al. have investigated into the efficacy of a diet used in Irritable Bowel Syndrome for the treatment of Small intestinal bacterial overgrowth.
SIBO has been observed in IBS and the first line of treatment is antibiotics. The problem using the antibiotics is that it destroys the bacteria indiscriminately. Instead using the antibiotics, the authors argue that the dietary intervention along with antibiotics might be a good option to target this bacterial overgrowth. They conducted a literature research about the studies investigated into dietary intervention such as Low FODMAP diet, probiotic food etc. on the bacterial growth and argued on both sides that it can or can not be used to alter the bacterial growth trajectory without affecting intestinal microbiota.
Although the manuscript is written in details with properly cited materials, following changes are also warranted:
1) The author should include some details about Low FODMAP diet in the introduction part.
2) There are a lot of typos in the manuscript which should be addressed. Following are the some of the examples:
Page,7 line 214, "inscreased" should be "increased"
Page,8 line 235, "Hydrolyzsed" should be "Hydrolysed" or Hydrolyzed"
Page 9, line 262, "Strenght" should be "Strength"
Page 9, line 270, "dietas" should be "diets"
Author Response
Dear Reviewer,
Thank you for all your work on our manuscript “Efficacy of an Irritable Bowel Syndrome diet in the treatment of Small Intestinal Bacterial Overgrowth: A Narrative Review”. Your comments and suggestions were very useful
and helped to improve the paper considerably. All your suggestions have been taken into account in the recent revision of the manuscript. You can find answers to your specific comments below:
- We have added essential information about Low FODMAP diet in the introduction section.
- Manuscript has been corrected and carefully checked by one of the Author who is a native speaker- Nicole Domanski
- All of typos have been corrected

Reviewer 2 Report
The topic undertaken by the authors is important and attractive. However, a few corrections have to be made, so the paper is easier to read and more informative.
Lines 130-146: The FODMAP foods/diet (low/high) and its importance in SIBO patients should be explained at the beginning of the section. The last paragraph of 3.1 should mostly summarize/maybe discuss a bit the diet’s pros and cons with reviewed reports. The same with the following sections, some introduction to the topic should be included (FODMAP/Monoprobiotics/Fiber), as it is in the 3.4 section – or enrich the Introduction section, where the FODMAP is firstly introduced but not explained and add fiber, and monoprobiotics. In the present form, the text is hard to read.
Tables 2-4 are not cited in the text
Table 3 – improve the arrows
Line 217 – citation 57 is incorrect, and there should be inulin, not insulin
The text needs revision – spelling mistakes, double spaces, missing commas, etc.
Author Response
Dear Reviewer,
Thank you for all your work on our manuscript “Efficacy of an Irritable Bowel Syndrome diet in the treatment of
Small Intestinal Bacterial Overgrowth: A Narrative Review”. Your comments and suggestions were very useful
and helped to improve the paper considerably. All your suggestions have been taken into account in the
recent revision of the manuscript. You can find answers to your specific comments below:
- We have corrected the text to make our manuscript easier to read. We realized that enriching the introduction with information about the Low FODMAP diet, fiber and probiotics (monoprobiotics) made the text more informative. Due to this fact, we decided to only summarize the Discussion part.
- It has been corrected and carefully checked by one of the Author who is a native speaker- Nicole Domanski.

Reviewer 3 Report
In the present review article Authors discussed whether diets specific for IBS may be helpful for SIBO.
My most relevant criticism is that most of discussion is merely speculative, since it is based on the hypothesis that, since IBS and SIBO may share some common features, a diet which is effective for IBS may be beneficial for SIBO as well. However, this approach is not evidence-based. Indeed, as Authors themselves affirmed (page 3 lines 103-104), “none of the studies evaluated the impact of this diet in SIBO patients”. Therefore, the utility to perform a review about this topic is questionable.
Author Response
Dear Reviewer,
Thank you very much for your feedback. We agree with what you have indicated in your comments. However, the main reason why we decided to write this manuscript was to show that the information and recommendations available for the SIBO diet are mainly extensions of the data from IBS. Duplicating IBS recommendations in patients with SIBO is increasingly used in clinical practice, however, as we now see, there is no evidence to confirm these recommendations in SIBO patients.
In our opinion, the information presented in this manuscript is significant and emphasizes the need for future research on this topic, in order to develop evidence-based recommendations for the SIBO diet.
For example, currently the main basis of the SIBO diet is to eliminate or alleviate SIBO gastrointestinal symptoms, but does not consider modulating the gut microbiota. As was included in our manuscript, some interventions like a prolonged elimination diet may cause negative changes in microbiome (gut microbiome dysbiosis). Others, like supplementing with monoprobiotics, fiber and focusing on mindful eating, might enhance SIBO treatment. Moreover, this is the first review including only monoprobiotic strains, which demonstrated a need for careful selection of bacterial species, with proper nomenclature.

Round 2
Reviewer 2 Report
The authors have corrected the manuscript according to the review. I recommend the manuscript for publication.
Reviewer 3 Report
Answer was not satisfactory. The article is not evidence based. My opinion i still rejection.